Mental practice in isolation improves cervical joint position sense in patients with chronic neck pain: a randomized single-blind placebo trial

Cuenca-Martínez Ferran 1 2
La Touche Roy Roylatouche@yahoo.es 1 2 3 4
León-Hernández Jose Vicente 1 2
Suso-Martí Luis 2 5
1 Departamento de Fisioterapia, Centro Superior de Estudios Universitarios La Salle, Universidad Autónoma de Madrid , Madrid , Spain
2 Motion in Brains Research Group, Institute of Neuroscience and Sciences of the Movement (INCIMOV), Centro Superior de Estudios Universitarios La Salle, Universidad Autónoma de Madrid , Madrid , Spain
3 Instituto de Dolor Craneofacial y Neuromusculoesquelético (INDCRAN) , Madrid , Spain
4 Instituto de Investigación Sanitaria del Hospital Universitario La Paz (IdiPAZ) , Madrid , Spain
5 Department of Physiotherapy, Universidad Cardenal Herrera-CEU, CEU Universities , Valencia , Spain
Keogh Justin
Electronic publication date: 2019 Sep 12
Publication date: 2019
Volume: 7
Electronic Location ID: e7681
Received 2019 Jul 25; Accepted 2019 Aug 16
Copyright: ©2019 Cuenca-Martínez et al.
Copyright year: 2019
Copyright holder: Cuenca-Martínez et al.
License: This is an open access article distributed under the terms of the Creative Commons Attribution License, which permits unrestricted use, distribution, reproduction and adaptation in any medium and for any purpose provided that it is properly attributed. For attribution, the original author(s), title, publication source (PeerJ) and either DOI or URL of the article must be cited.
License URL: https://creativecommons.org/licenses/by/4.0/

Keywords: Action observation, Motor imagery, Joint position sense, Chronic neck pain

Funding: The authors received no funding for this work.

==============================
Objective

The main objective of this trial was to assess whether action observation (AO) training and motor imagery (MI) produced changes in the cervical joint position sense (CJPS) both at the end of the intervention and 10 min postintervention compared with a placebo intervention in patients with nonspecific chronic neck pain (NSCNP).

Methods

A single-blind placebo clinical trial was designed. A total of 30 patients with NSCNP were randomly assigned to the AO group, MI group or placebo observation (PO) group. CJPS in flexion, extension and rotation movements in both planes were the main variables.

Results

The results obtained in the vertical plane showed that the AO group obtained greater improvements than the PO group in the CJPS in terms of cervical extension movement both at the end of the intervention and 10 min postintervention (p = .001, d = 1.81 and p = .004, d = 1.74, respectively), and also in cervical flexion movement, although only at 10 min after the intervention (p = .035, d = 0.72). In addition, the AO group obtained greater improvements than the MI group in the CJPS only at the end of the intervention in cervical extension movement (p = .041, d = 1.17). Regarding the left rotation cervical movement, both the MI and AO groups were superior to the PO group in both planes at the end of the intervention (p < .05, d > 0.80).

Conclusions

Although both AO and MI could be a useful strategy for CJPS improvement, the AO group showed the strongest results. The therapeutic potential of the application of mental practice in a clinical context in the early stages of rehabilitation of NSCNP should be considered.

Introduction

Neck pain is a common musculoskeletal disorder with a high prevalence in the population and is the fourth leading disability-generating condition (Vos et al., 2013). Chronic neck pain is often considered nonspecific (NSCNP), due to the difficulty in identifying the origin of the pain, when imaging tests provide no relevant information for establishing an accurate pathological diagnosis (Bogduk, 2011). This clinical condition is thought to have a multidimensional nature due to the combination of a complex pathogenesis, the presence of maladaptive processes of central neuroplasticity and in pain processing, as well as the relevance of psychological aspects involved in the NSCNP such as anxiety or pain catastrophism (Binder, 2007; Dimitriadis et al., 2015; Muñoz García et al., 2016).

It is commonly reported that patients with NSCNP present an alteration in the cervical joint position (Alahmari et al., 2017). The cervical region has a large number of proprioceptive receptors, especially in the upper cervical spine (Falla, Jull & Hodges, 2004; Falla, Bilenkij & Jull, 2004). It has been suggested that in patients with NSCNP, proprioceptive afferent information from the cervical spine might be impaired due to the presence of chronic pain (Uremović et al., 2007). In addition, Kim, Kim & Nabekura (2017) have suggested that patients with persistent pain might undergo a process of maladaptive neuroplasticity in major sensitive areas such as the primary somatosensory area. Furthermore, Hodges & Tucker (2011) suggested that maladaptive processes of central plasticity could lead to impaired motor planning and movement execution as a pain response and could therefore affect motor control and movement acuity in this region.

To improve proprioceptive input in the cervical region, several interventions have been proposed, including craniocervical motor control exercises (MCEs) (Izquierdo et al., 2016; Kim & Kwag II, 2016; Lee & Kim, 2016). MCEs have been shown to reduce pain and disability in patients with NSCNP compared with other types of treatments (Martin-Gomez et al., 2019). However, it has been suggested that patterns of muscle activation and recruitment are altered in the presence of pain, so implementing MCEs is a challenging aspect in these patients and might lead patients to perform them incorrectly, which could reduce their effectiveness (Sterling, Jull & Wright, 2001a). One of the alternatives in those early stages of intervention could be mental practice based on mental motor imagery (MI) and action observation (AO).

MI is defined as a dynamic mental process that involves the representation of an action, in an internal manner, without its actual motor execution (Decety, 1996). AO evokes an internal, real-time simulation of what the observer is seeing (Buccino, 2014). It has been shown that both MI and AO training can activate neurocognitive mechanisms underlying the planning and execution of voluntary movements in a similar manner as when this movement is actually performed (Wright, Williams & Holmes, 2014). Previously Villafañe et al. (2016) found an improvement in motor function through mental practice after total hip arthroplasty.

The authors hypothesize that both forms of mental practice in isolation could lead to changes in cervical joint position sense (CJPS) compared with a placebo intervention. Therefore, the main objective of the present study was to assess whether AO training and mental MI produced changes in CJPS both at the end of the intervention and 10 min postintervention in comparison with a placebo intervention in patients with NSCNP.

Methods

Study design

The present study was a randomized, single-blind placebo clinical trial, planned and conducted in accordance with Consolidated Standards of Reporting Trials requirements (Schulz et al., 2010) (Fig. 1) and was approved by La Salle University Center for Higher Education (CSEULS-PI-027/2019).

Figure 1 Study flow chart.

This study was registered in the United States Randomized Trials Registry on https://www.clinicaltrials.gov/ (trial registry number: NCT03910829). All patients completed the informed consent document prior to the study.

Recruitment of participants

Patients who were diagnosed by their family doctor as having NSCNP were referred to the primary care physiotherapy service, and all met the inclusion criteria of the study at one physiotherapy center. Participants were recruited between April 2019 and May 2019.

The inclusion criteria were as follows: (a) men and women aged between 18 and 65 years; and (b) a medical diagnosis of NSCNP with at least 6 months of neck pain symptoms. Exclusion criteria were the following: (a) patients with rheumatic diseases, cervical hernia or radicular pain, cervical whiplash syndrome, neck surgeries or a history of arthrodesis; (b) systemic diseases; (c) vision, hearing or vestibular problems; and (d) severe trauma or a traffic accident that had an impact on the cervical area. All the participants were given an explanation of the study procedures, which were planned under the ethical standards of the Helsinki Declaration. In addition, this randomized, single-blind placebo clinical trial was used as a pilot with the aim of calculating the sample size of a future study.

Randomization

Randomization was performed using a computer-generated random sequence table with a non-balanced 3-block design (GraphPad Software, Inc., CA, USA). An independent researcher generated the randomization list, and a member of the research team who was not involved in the assessment or intervention of the participants was in charge of the randomization and maintained the list. Those included were randomly assigned to 1 of the 3 groups using the random-sequence list, ensuring concealed allocation.

Blinding

The assessments and treatments were performed by different therapists. The evaluator was blinded to the participant’s assignment. All the intervention procedures were performed by the same physiotherapist, who had more than three years of experience in the field and was blinded to the purpose of the study. Patients were blinded to their group allocation.

Interventions

The interventions were previously described by Suso-Martí et al. (2019).

Action observation group

Patients in this group performed an exclusive AO protocol of 2 commonly used MCEs in the treatment of patients with NSCNP (Jull et al., 2009; Jull et al., 2007; O’Leary et al., 2007). Both exercises were based on the motor gesture of craniocervical flexion (Fig. 2). Patients in the AO group performed the observation by watching a video of the continuous performance of both exercises repeatedly during 2 series of 1 min for each exercise, with a total duration of 4 min.

Figure 2 Protocol of the intervention.

The first exercise consisted of maintaining the cervical spine in a neutral position in a sitting position and performing a deep muscle contraction to flatten the curve of the neck, nodding with the head. The second exercise involved a deep muscle contraction by performing the craniocervical flex-extension gesture with the resistance of an elastic band.

Motor imagery group

The patients in this group performed an MI protocol of the same cervical exercises as the AO group (Fig. 2). Patients were instructed on the movements they were to imagine by showing both exercises and the precise instructions for each movement. After this, they were instructed to perform a third-person mental task of visual MI of both exercises during 2 series of 1 min for each exercise, with a total duration of 4 min.

Placebo observation group

Patients in the placebo observation (PO) group underwent a placebo AO protocol. The patients watched during the same intervention time as both previous groups. This documentary video was composed of video clips of nature landscapes, without any human agent or motor gesture.

This type of placebo AO protocol has been used in previous studies (Bang et al., 2013; Buccino et al., 2012).

Outcomes

Primary outcomes

Cervical joint position sense.

CJPS is an objective measure of neck repositioning sense and can quantify the alteration in neck proprioception. CJPS was assessed with a visual feedback device, the SenMoCOR LED (Sensory Motor Control-Oriented Rehabilitation, IAOM-US). This device consists of adjustable straps and a fastening support for a laser beam. It is adjustable to the evaluator’s desired position, allowing projection of the light beam.

The experimental procedure for assessing CJPS with a laser beam has been described by Revel, Andre-Deshays & Minguet (1991). First, patients were asked to sit in a comfortable position at 90-cm distance from the bullseye with the SenMoCOR Kit correctly placed. With eyes closed, they were asked to point to the neutral position of the head and memorize it to return to after the completion of the movement. This point was recorded as a reference for each patient. The patient subsequently performed a maximal movement of cervical flexion and then attempted to find the initial reference position with a maximum of accuracy without speed instructions. The point on which the light beam stopped indicated the global error measured in centimeters (cm) in relation to the center of the target recorded previously. The assessor measured the deviations from de target position for each trial on both axes (x∕y). Values of x (abscissa) and y (ordinate) were recorded according to the Cartesian coordinate system. The same protocol was used for the extension, right and left rotation movements. Ten trials were performed with head repositioning after each movement, and the mean measure was recorded. No feedback was given to the participants about their actual performance. This CJPS testing method offers an easy, quick and inexpensive method for measuring cervical joint position sense. The test presents inter-rater reliability ranging from moderate to good/substantial agreement (intraclass correlation coefficient [ICC] ≥ 0.51–0.75); it also presents intra-rater reliability ranging from moderate to almost perfect agreement (ICC ≥ 0.48–0.82) (Juul et al., 2013). The minimal detectable change ranged from 0.52–0.75 cm (Juul et al., 2013).

Secondary outcomes

Ability to generate mental motor imagery.

The movement imagery questionnaire-revised (MIQ-R) is an 8-item self-report inventory that was used to assess visual and kinesthetic motor imagery ability. Four different movements are included in the MIQ-R, which is comprised of four visual and four kinesthetic items. For each item, participants read a description of the movement. They then physically performed the movement and were instructed to reassume the starting position after finishing the movement and before performing the mental task, imagining the movement visually or kinesthetically. Each participant then rated the ease or difficulty of mentally generating that image on a 7-point scale, in which 7 indicates “very easy to see/feel” and 1 “very difficult to see/feel”. The internal consistencies of the MIQ-R have been consistently adequate, with Cronbach’s α coefficients ranging above 0.84 for the total scale, 0.80 for the visual subscale and 0.84 for the kinesthetic subscale (Campos & González, 2010).

Mental chronometry.

Mental chronometry (MC) is a reliable measure that has been widely used to record objective measurements of the ability to create mental motor images (Guillot & Collet, 2005; Malouin et al., 2008; Williams et al., 2015). To assess MC, first, the time dedicated to imagining each task was recorded using a stopwatch. The time between the interval command to start the task (given by the evaluator) and the verbal response at the conclusion of the task (given by the participant) was recorded. After the MI task, the participants were asked to perform the real movement execution of the task, and the time dedicated to performing each task was recorded using a stopwatch. Both time measurements were taken to obtain the temporal congruence between the tasks. In healthy participants, for the temporal congruence test, the ICC ranged from 0.63 to 0.95, whereas the ICC for intrasession reliability ranged from 0.95 to 0.97 (Malouin et al., 2008).

Pain-related fear of movement.

Pain-related fear of movement was assessed using the 11-item Spanish version of the Tampa Scale of kinesiophobia, whose reliability and validity have been demonstrated (Gómez-Pérez, López-Martínez & Ruiz-Párraga, 2011). The Tampa scale for Kinesiophobia consists of 2 subscales, one related to fear of activity and the other related to fear of harm. The final score can range between 11 and 44 points, with higher scores indicating greater perceived kinesiophobia (Gómez-Pérez, López-Martínez & Ruiz-Párraga, 2011). Internal consistency ratings were moderate. In the chronic pain sample, Cronbach’s α = 0.79 was obtained using total TSK items (Gómez-Pérez, López-Martínez & Ruiz-Párraga, 2011).

Neck disability.

Disability was measured using the Spanish-validated Neck Disability Index (NDI), which consists of 10 items related to daily functional activities. Each question is measured on a scale from 0 (no disability) to 5, and an overall score out of 100 is calculated by adding each item score together and multiplying it by 2. A higher NDI score indicates greater perceived disability due to neck pain. It has been shown to have high “test-retest” reliability and to have appropriate psychometric properties (Alfonso Andrade Ortega, Damián Delgado Martínez & Almécija Ruiz, 2008). MacDermid et al. (2009) concluded that differences in 7 points out of 50 in the NDI should be considered as clinically relevant. In addition, intraclass correlation coefficients (ICCs) ranged from 0.50 to 0.98.

Level of physical activity.

The level of physical activity was assessed using the IPAQ questionnaire, which allows the participants to be divided into 3 groups according to their level of activity, which can be high, moderate, and low or inactive (Roman-Viñas et al., 2010). This questionnaire has shown acceptable validity and psychometric properties to measure total physical activity. Therefore, the psychometric properties of the questionnaire were accepted for use in studies that required the measurement of physical activity; reliability was approximately 0.65 (r = 0.76; 95% CI [0.73–0.77]) (Mantilla Toloza & Gómez-Conesa, 2007).

Procedures

Data were collected as previously described by Suso-Martí et al. (2019). Each participant was given an informed consent document to participate in the study, in addition to a set of questionnaires to complete before starting the intervention. These questionnaires included psychometrics forms and a questionnaire about age, gender, time with pain duration and pain intensity. The psychological variables were evaluated with self-assessments. Then MIQ-R and mental chronometry were assessed, and the pre-intervention measurements of CJPS were then taken. Subsequently, in a sitting position, patients performed the AO, MI or PO protocol, according to their group. Immediately after the intervention, a blinded evaluator measured the CJPS in the four movements. Following this, patients were asked to sit and relax comfortably, without movement, for 10 min, and the CJPS was measured again (post 2).

Statistical analysis

The statistical analysis was performed using SPSS software version 22.0 (SPSS Inc., Chicago, IL, USA).

The normality of the variables was evaluated using the Shapiro–Wilk test. Descriptive statistics were used to summarize the data for the continuous variables and are presented as mean ± standard deviation and 95% confidence interval. A two-way repeated measures analysis of variance (ANOVA) was conducted to study the effect of the between-subject factor ‘intervention group’ with three categories (i.e., AO, MI and PO) and the within-subject called ‘time’ with also three categories (i.e., pre, post, and post 2) on the dependent variables. Partial eta squared (ηp2) was calculated as a measure of effect size (strength of association) for each main effect and interaction in the ANOVAs, with 0.01–0.059 representing a small effect, 0.06–0.139 a medium effect and >0.14 a large effect (Cohen, 1973). A post hoc analysis with Bonferroni correction was performed in the case of significant ANOVA findings for multiple comparisons between variables. Effect sizes (d) were calculated according to Cohen’s method, in which the magnitude of the effect was classified as small (0.20–0.49), moderate (0.50–0.79) or large (0.8) (Cohen, 1988). The α level was set at .05 for all tests. Additionally, we compared age, weight and height between groups, to explore whether the groups were homogeneous at baseline with a 1-factor ANOVA.

Results

A total of 30 patients with NSCNP were included and were randomly allocated into three groups of 10 participants per group. There were no adverse events reported in either group. No statistically significant differences in demographic data were present preintervention between the groups and the self-report variables (Table 1).

Table 1 Descriptive statistics of sociodemographic, self-reported and psychosocial data.

Measures	AO (n = 10)	MI (n = 10)	PO (n = 10)	p value	
Age	33.5  ± 14.25	30.6  ± 11.53	27.70  ± 6.39	.520	
Height (cm)	171.9  ± .80	123.10  ± .70	174  ± .40	.798	
Weight (Kg)	66.7  ± 7.97	68.70  ± 4.8	69.5  ± 8.26	.672	
VAS	68.9  ± 13.95	75  ± 7.73	70.8  ± 9.36	.437	
Pain duration (m)	27.9  ± 17.99	26.2  ± 12.45	17.4  ± 10.05	.212	
TSK-11	32.3  ± 6	33  ± 4.85	31.3  ± 3.93	.633	
NDI	30.5  ± 3.62	29.8  ± 3.82	32.1  ± 4.48	.430	
IPAQ	1760.6  ± 483.51	1713.85  ± 500.3	1785.7  ± 659.17	.958	
MIQ-R	47.4  ± 4.77	47.3  ± 7.86	48  ± 4.52	.960	
MC	3.65  ± 3.96	4.39  ± 5.7	4.71  ± 4.52	.879	
Sex				.875	
Male	5 (50)	5 (50)	4 (40)		
Female	5 (50)	5 (50)	6 (60)		
Educational level				.03	
Secondary education	3 (30)	5 (50)	0 (00)		
College education	7 (70)	5 (50)	10 (100)		
Marital status				.136	
Single	7 (70)	3 (30)	5 (50)		
Married	3 (30)	4 (40)	4 (40)		
Divorced	0 (0)	3 (30)	1 (0)		
Medication				.563	
Yes	7 (70)	5 (50)	7 (70)		
No	3 (30)	5 (50)	3 (30)		
Pain Location				.530	
Right	5 (50)	2 (20)	2 (20)		
Left	3 (30)	5 (50)	4 (40)		
Both	2 (20)	3 (30)	4 (40)		
Notes.

Values are presented as mean  ± standard deviation or number (%).

MI motor imagery

AO action observation

PO placebo observation group

TSK Tampa Scale of Kinesiophobia

NDI neck disability index

IPAQ International Physical Activity Questionnaire

MIQ-R Movement Imagery Questionnaire-Revised

MC mental chronometry

VAS visual analog scale

Flexion range of motion

X-plane

Regarding the flexion range of motion (ROM) in the X-plane, the ANOVA revealed significant changes in group*time (F = 4.06, p = .006, ηp2=0.231) and time (F = 17.45, p < .001, ηp2=0.393). The post hoc analysis revealed significant within-group differences in both the MI and AO groups, with a moderate-large effect size for the MI group at postintervention (p < .001, d = 0.95) and at post 2 intervention (p = .021, d = 0.72), as well as with a moderate-large effect size for AO at postintervention (p < .001, d = 0.96) and at post 2 intervention (p = .001, d = 0.74). The post hoc analysis revealed no significant within-group differences in the PO group (p > .05) (Table 2). However, no significant differences were found between the groups (p > .05).

Table 2 Within-group differences in flexion cervical movement.

Measure	Group	Mean± SD	Mean difference (95% CI); Effect size (d)	
		Pre	Post	Post 2	(a) pre–post	
		(b) pre–post 2	
FlexionX-plane	PO	12.0 ± 4.3	11.3  ± 4.4	12.16  ± 6.1	(a) 0.7 (−2.1 to 3.5); d= 0.16	
(b) −0.1 (−4.0 to 3.8); d =  − 0.1	
MI	14.6 ± 6.0	9.2  ± 5.3	10.14  ± 6.4	(a) 5.4** (2.5–8.2); d= 0.95	
(b) 4.4* (0.5–8.4); d= 0.72	
AO	12.6 ± 8.74	6.7  ± 7.3	6.1  ± 4.1	(a) 5.9** (3.1–8.8); d= 0.96	
(b) 6.5* (2.6–10.4); d= 0.74	
Measure	Group	Mean± SD	Mean difference (95% CI); Effect size (d)	
		Pre	Post	Post 2	(a) pre–post	
		(b) pre–post 2	
Flexion  Y-plane	PO	9.9 ± 4.3	8.9 ± 5.4	10.8 ± 4.0	(a) 0.9 (−1.6 to 3.4); d= 0.20	
(b) −0.9 (−3.6 to 1.8); d =  − 0.21	
MI	8.8 ± 3.6	7.6 ± 4.0	7.4 ± 4.4	(a) 1.2 (−1.3 to 3.7); d= 0.31	
(b) 1.4 (−1.3 to 4.1); d= 0.34	
AO	9.4 ± 5.4	4.6 ± 3.7	5.6 ± 4.3	(a) 4.7** (2.2–7.2); d= 1.01	
(b) 3.8* (1.0–6.5); d= 0.72	
Notes.

* p < .05.

** p < .001.

CI confidence interval

SD standard deviation

PO placebo observation group

MI motor imagery group

AO action observation group

Y-plane

Regarding the flexion ROM in the Y-plane, the ANOVA revealed significant changes in group*time (F = 4.14, p = .005, ηp2=0.235) and time (F = 8.83, p < .001, ηp2=0.246). The post hoc analysis revealed significant within-group differences only in the AO group, with a moderate-large effect size at postintervention (p < .001, d = 1.01) and at post 2 intervention (p = .005, d = 0.77). The post hoc analysis revealed no significant within-group differences in the PO or MI groups (p > .05) (Table 2). Regarding the between groups comparison, only the AO group showed significant differences with the PO group at post 2 intervention, with a moderate effect size (p = .035, d = 0.72) (Fig. 3).

Figure 3 Between-group differences in flexo-extension cervical movements (Y-plane).

Extension range of motion

X-plane

Regarding the extension ROM in the X-plane, there were no significant differences in time (F = 1.87, p = .16, ηp2=0.065) or in group*time (F = 0.33, p = .33, ηp2=0.024).

Y-lane

Regarding the extension ROM in the Y-plane, the ANOVA revealed significant changes in group*time (F = 6.87, p < .001, ηp2=0.337) and time (F = 8.56, p = .001, ηp2=0.241). The post hoc analysis revealed significant within-group differences in both the MI and AO groups, with a moderate-large effect size for MI at postintervention (p = .017, d = 0.77) and at post 2 intervention (p = .007, d = 1.04), as well as with a large effect size for AO at postintervention (p < .001, d = 1.01) and at post 2 intervention (p = .006, d = 1.05). The post hoc analysis revealed no significant within-group differences in the PO group (p > .05) (Table 3).

Table 3 Within-group differences in extension cervical movement.

Measure	Group	Mean± SD	Mean difference (95% CI); Effect size (d)	
		Pre	Post	Post 2	(a) pre–post	
		(b) pre–post 2	
ExtensionX-plane	PO	11.0 ± 3.8	10.1  ± 2.8	10.6  ± 5.6	(a) 0.9 (−2.7 to 4.6); d= 0.27	
(b) 0.4 (−3.2 to 4.0); d= 0.08	
MI	9.1 ± 4.0	7.6  ± 3.3	8.8  ± 3.6	(a) 1.4 (−2.2 to 5.2); d= 0.41	
(b) 0.3 (−3.3 to 3.9); d= 0.07	
AO	9.0 ± 5.5	6.9  ± 3.7	6.8  ± 3.7	(a) 2.1 (−1.5 to 5.8); d= 0.44	
(b) 2.2 (−1.3 to 5.7); d= 0.47	
Measure	Group	Mean± SD	Mean difference (95% CI); Effect size (d)	
		Pre	Post	Post 2	(a) pre–post	
		(b) pre–post 2	
ExtensionY-plane	PO	11.1 ± 2.6	13.3 ± 4.1	12.6 ± 4.0	(a) −2.2 (−4.9 to 0.5); d =  − 0.64	
(b) −1.4 (−5.0 to 2.1); d =  − 0.44	
MI	13.7 ± 4.3	10.5 ± 4.0	8.9 ± 4.7	(a) 3.2* (0.5–5.9); d= 0.77	
(b) 4.7* (1.1–8.3); d= 1.04	
AO	10.7 ± 5.8	5.4 ± 4.5	5.9 ± 2.8	(a) 5.2** (2.5–8.0); d= 1.01	
(b) 4.8* (1.2–8.4); d= 1.05	
Notes.

* p < .05.

** p < .001.

CI confidence interval

SD standard deviation

PO placebo observation group

MI motor imagery group

AO action observation group

Regarding the between groups comparison, the AO group showed significant differences with both the MI and the PO groups at postintervention, with a large effect size (p = .041, d = 1.17, and p = .001, d = 1.81, respectively) and at post 2 intervention with only the PO group, with a large effect size ( p = .004, d = 1.74) (Fig. 3).

Left rotation range of motion

X-plane

Regarding the left rotation ROM in the X-plane, the ANOVA revealed significant changes in group*time (F = 3.08, p = .023, ηp2=0.186) but not in time (F = 1.53, p = .226, ηp2=0.054). The post hoc analysis revealed no significant within-group differences in any group (p > .05) (Table 4). However, both the MI and AO groups showed significant between group differences with the PO group at postintervention, with a large effect size (p = .035, d = 1.29, and p = .005, d = 1.54, respectively) (Fig. 4).

Table 4 Within-group differences in left rotation cervical movement.

Measure	Group	Mean± SD	Mean difference (95% CI); Effect size (d)	
		Pre	Post	Post 2	(a) pre–post	
		(b) pre–post 2	
Left rotation  X-plane	PO	11.0 ± 4.8	12.8  ± 3.7	12.9  ± 3.6	(a) −1.8 (−4.9 to 1.2); d =  − 0.42	
(b) −1.9 (−4.2 to 0.4); d =  − 0.44	
MI	9.3 ± 4.1	7.8  ± 4.1	8.4  ± 3.4	(a) 1.5 (−1.6 to 4.6); d= 0.36	
(b) 0.9 (−1.4 to 3.3); d= 0.24	
AO	9.3 ± 5.3	6.3  ± 4.7	8.6  ± 4.7	(a) 2.9 (−0.1 to 6.0); d= 0.59	
(b) 0.7 (−1.6 to 3.1); d= 0.14	
Measure	Group	Mean± SD	Mean difference (95% CI); Effect size (d)	
		Pre	Post	Post 2	(a) pre–post	
		(b) pre–post 2	
Left rotationY-plane	PO	10.5 ± 3.6	12.6  ± 5.3	9.7  ± 2.3	(a) −2.1 (−6.3 to 2.0); d =  − 0.46	
(b) 0.7 (−2.1 to 3.6); d= 0.26	
MI	12.3 ± 5.1	7.1  ± 3.2	8.6  ± 2.8	(a) 5.1* (1.0–9.3); d= 1.18	
(b) 3.6* (0.7–6.5); d= 0.84	
AO	11.0 ± 4.4	5.1  ± 3.3	6.4  ± 3.6	(a) 5.8* (1.6–10.0); d= 1.71	
(b) 4.5* (1.7–7.4); d= 1.25	
Notes.

* p < .05.

** p < .001.

CI confidence interval

SD standard deviation

PO placebo observation group

MI motor imagery group

AO action observation group

Figure 4 Between-group differences in rotation cervical movements (X-plane).

Y-plane

Regarding the left rotation ROM in the Y-plane, the ANOVA revealed significant changes in group*time (F = 5.44, p = .002, ηp2=0.287) and time (F = 9.58, p = .001, ηp2=0.262). The post hoc analysis revealed significant within-group differences in both the MI and AO groups, with a large effect size for MI at postintervention (p = .012, d = 1.18) and at post 2 intervention (p = .009, d = 0.87), as well as with a large effect size for AO at postintervention (p = .004, d = 1.71) and at post 2 intervention (p = .001, d = 1.25). The post hoc analysis revealed no significant within-group differences in the PO group (p > .05) (Table 4). In addition, both the MI and AO groups showed significant between group differences with the PO group at postintervention, with a large effect size (p = .016, d = 1.24, and p = .001, d = 1.70, respectively) (Fig. 5).

Figure 5 Between-group differences in rotation cervical movements (Y-plane).

Right rotation range of motion

X-plane

Regarding the right rotation ROM in the X-plane, the ANOVA revealed significant changes in group*time (F = 2.81, p = .034, ηp2=0.172) but not in time (F = 1.98, p = .147, ηp2=0.069). The post hoc analysis revealed significant within-group differences in the AO group, with a moderate effect size, only at postintervention (p = .011, d = 0.72). However, significant within-group differences were also found between post-post2 intervention in the AO group, showing a loss of effect after 10 min (p = .02, d =  − 0.61) (Table 5). Finally, no significant differences were found between the groups (p > .05) (Fig. 4).

Table 5 Within-group differences in right rotation cervical movement.

Measure	Group	Mean± SD	Mean difference (95% CI); Effect size (d)	
		Pre	Post	Post 2	(a) pre–post	
		(b) pre–post 2	
Right rotation  X-plane	PO	10.1 ± 4.3	11.0  ± 4.1	10.4  ± 5.1	(a) −0.9 (−3.8 to 2.0); d =  − 0.21	
(b) −0.3 (−3.6 to 3.0); d =  − 0.06	
MI	9.6 ± 3.6	8.3  ± 3.8	7.5  ± 4.1	(a) 1.2 (−1.6 to 4.2); d= 0.35	
(b) 2.1 (−1.2 to 5.4); d= 0.54	
AO	10.3 ± 5.6	6.7  ± 4.5	9.8  ± 5.4	(a) 3.6* (0.7–6.5); d= 0.72	
(b) 0.5 (−2.7 to 3.8); d= 0.09	
Measure	Group	Mean± SD	Mean difference (95% CI); Effect size (d)	
		Pre	Post	Post 2	(a) pre–post	
		(b) pre–post 2	
Right rotation  Y-plane	PO	9.5 ± 3.3	7.5  ± 2.7	10.4  ± 4.0	(a) 2.0 (−2.3 to 6.4); d= 0.66	
(b) −0.8 (−5.2 to 3.5); d =  − 0.24	
MI	9.8 ± 4.4	7.4  ± 2.6	8.9  ± 3.4	(a) 2.4 (−1.9 to 6.7); d= 0.66	
(b) 0.9 (−3.4 to 5.2); d= 0.22	
AO	11.7 ± 5.4	5.9  ± 3.7	7.3  ± 3.8	(a) 5.8* (1.5–10.2); d= 1.24	
(b) 4.4* (0.1–8.8); d= 0.94	
Notes.

* p < .05.

CI confidence interval

SD standard deviation

PO placebo observation group

MI motor imagery group

AO action observation group

Y-plane

Regarding the right rotation ROM in the Y-plane, the ANOVA revealed significant changes over time (F = 7.53, p = .003, ηp2=0.218) but not in group*time (F = 1.75, p = .151, ηp2=0.115). The post hoc analysis revealed significant within-group differences only in the AO group, with a large effect size, at postintervention (p = .006, d = 1.24) and at post 2 intervention (p = .043, d = 0.94). The post hoc analysis revealed no significant within-group differences in the PO or MI groups (p > .05) (Table 5). Finally, no significant differences were found between the groups (p > .05) (Fig. 5).

Sample size calculation

The sample size was estimated with the program G*Power 3.1.7 for Windows (G*Power from University of Dusseldorf, Germany) (Faul et al., 2007). The sample size calculation was considered as a power calculation to detect between-group differences in a primary outcome measures (flexion movement). We considered three groups and two measurements for primary outcomes to obtain 95% statistical power (1- β error probability) with an α error level probability of 0.05 using analysis of variance (ANOVA) of repeated measures, within-between interaction, and an effect size of  ηp2 = 0.231 obtained from our results. This generated a sample size of total of 42 participants plus an estimated 15% loss in follow-up, yielding a total of 48 participants (16 per group).

Discussion

The main objective of the present study was to assess whether AO training and mental MI produced changes in CJPS both at the end of the intervention and at 10 min postintervention compared with a placebo intervention in patients with NSCNP.

The results obtained in the vertical plane showed that the AO group obtained greater improvements than the PO group in CJPS of the cervical extension movement both at the end of the intervention and at 10 min postintervention, as well as in the cervical flexion movement, although only at 10 min after the intervention. In addition, the AO group obtained greater improvements than the MI group in CJPS only at the end of the intervention of the cervical extension movement. However, in the horizontal plane of the flexo-extension movements, neither of the two mental practice groups was superior to the placebo intervention.

On the other hand, the results obtained in the vertical plane showed that both the AO and MI groups obtained greater improvements than the PO group in CJPS of the cervical left rotation movement at the end of the intervention. However, no significant differences were found between the groups in the right cervical rotation movement. Finally, in the horizontal plane, again both the AO and MI groups obtained greater improvements than the PO group in CJPS of the cervical left rotation movement at the end of the intervention, but no significant differences were found in the right cervical rotation movement between the groups.

NSCNP usually presents an alteration in CJPS (Alahmari et al., 2017). Chronic pain could affect receptors and the transmission of proprioceptive information from the cervical region, one of the keys to an adequate sense of joint position (Uremović et al., 2007). In addition, impaired transmission of proprioceptive information in patients with chronic musculoskeletal pain could lead to neuroplastic reorganization of body schema in the primary somatosensory area (Kim, Kim & Nabekura, 2017). Body schema is the model that is used by the musculoskeletal system for control, and its disruption is thought to cause incongruence between motor output and proprioceptive feedback (Mccormick et al., 2007). Therefore, in the current treatment of patients with NSCNP, proprioceptive training has been proposed, with the aim of improving internal representation both through exercise and manual therapy, which could reduce pain and disability in these patients (Treleaven, 2008). It is possible that the overlap of neural processes between MI and AO with real movement execution could provoke similar effects to real proprioceptive stimulation (Hardwick et al., 2018). Our results are consistent with those obtained by Beinert et al. (2015) in which improvements were also found in CJPS after imagination or observation of CJPS task. It is important to note that the exercises selected in the present study were specific to cranio-cervical flexo-extension movement pattern, whereas in the study mentioned above the mental practice was specific to the outcome variable. There are important differences between both types of exercise, since the mental practice of the exercises used in this study could lead to the learning of a motor gesture used in the rehabilitation of patients with NSCNP, which could lead to a difference from a clinical point of view.

It should be noted that the two movements observed or imagined by the patients in the present study were very subtle, low-joint path, highly complex, and precise movements in cervical flexo-extension, and they were the two movements exclusively performed along only the vertical plane. The findings showed that the strongest improvements obtained in CJPS were in the OA group in the same plane and in the same flex-extension movements. However, in the horizontal plane, no differences were found between the groups. These low joint range motor gestures were used because it has been found that observing full cervical movements can cause a fear response associated with movements perceived as dangerous (La Touche et al., 2018).

To respond to this finding, the role of mirror neurons and their relationship with the recognition of actions should be analyzed. Rizzolatti, Fogassi & Gallese (2001) have established the “direct-matching hypothesis”. According to this hypothesis, the OA provokes an automatic activation in the observer of the same cerebral areas related to the planning and real execution of the observed action. Given the result of the activation of these neural substrates during the execution of the action is known, the observation allows the observer to understand what is being observed through a specific observation-execution matching mechanism. Perhaps due to the high complexity of craniocervical motor control gestures, along with the fact that only single-plane movements were observed and imagined, we hypothesize that the activation of neural substrates is related to the planning and execution of voluntary movements specific to that plane. This hypothesis would explain the improvements in the CJPS in the movements of the same plane but not in the horizontal plane. However, this hypothesis is only neurophysiological, because the patients’ brain activity could not be observed directly. In addition, the AO and MI groups obtained greater improvements than the PO group in CJPS in terms of cervical left rotation movement; however, this result was not maintained 10 min after the intervention (post 2). There might also be a nonspecific movement plane mechanism to explain this result. Previous research has shown that mental practice provides an internal position body reference, which improves the spatiotemporal control of the position of the body in space during a dynamic movement, a critical aspect in CJPS. It has been hypothesized that both MI and AO produce better integration of motor actions due to a better internal body reference despite the absence of real movement (Papadelis et al., 2007). It is possible that this better internal reference of the position of the head with respect to the body could explain the positive results obtained in the left rotation, despite the fact that the mental gestures performed were in another plane of movement. However, further research is needed on the specific and nonspecific mechanisms in motor outputs induced by AO and MI.

The motor control exercises selected could influence the differences found between AO and MI training. Motor control exercises are difficult to imagine due to the fact that they require motor learning of difficult and precise movements. Previous research has shown that movement complexity and familiarity are related to MI performance (Paris-Alemany et al., 2019). Therefore, this intervention group might have been influenced by the difficulty of the mental motor image creation of these exercises. In addition, MI is less effective in people with less ability to perform it (Patterson et al., 2006), and it is well known that patients with chronic pain have a decreased ability to create mental motor images, which could also have affected our results (Breckenridge et al., 2019). Thus, taking into account all these variables, significant mental effort is required, which the patients might not have been able to achieve (Cuenca-Martínez et al., 2018; Decety et al., 1991). Regarding mental practice intervention duration, a meta-analysis by Driskell, Copper & Moran (1994) proposed a MI intervention for approximately 20 min is ideal to obtain the maximum benefit from MI. Therefore, Hinshaw (1991) suggest that MI duration from 10 to 15 min was required for the optimal effect on performance. Perhaps the short intervention time was insufficient to solve these challenges. In addition, Gonzalez-Rosa et al. (2015) have shown that AO provokes greater activation of cortical areas during motor learning and induces better motor learning results in comparison with MI. Taube et al. (2015) showed that cortical activity was higher during the combination of AO and MI, so it is possible that best training effects should be expected when participants apply MI during AO.

Moreover, previous studies have shown that AO could activate in a more ecological way the mirror neuron system in front of the MI (Gatti et al., 2013). The reason for this difference is that the ventral premotor cortex receives visual inputs and could be more activated by actual visual input than by the absence of visual input or overt movement, as in the case of motor imagery (Rizzolatti & Luppino, 2001). In addition, during AO, the observer has a model who performs the action and in the correct context. In contrast, during motor imagery, the individual must rehearse the relevant motor representations and covertly perform the action, and this could be especially relevant in subjects with diminished imagery ability, as mentioned above (Gatti et al., 2013; Mulder, 2007). This could lead to better motor learning via AO of new and highly complex tasks (Mulder et al., 2004), such as those included in the exercises in this study.

Finally, it is important to note that other therapeutic options have been used to improve cervical motor control in patients with NSCNP. For example, Martín-Rodríguez et al. (2019) recently found that dry needling both inside and outside the myofascial trigger point in the sternocleidomastoid muscle led to improvements in cervical motor control. In addition, Sterling, Jull & Wright (2001b) found that spinal manual therapy provoked a decreased activity of the superficial flexor muscle of the neck in a cervical motor control test. Therefore, in future studies, it would be interesting to compare the effect of mental practice against or even in combination with these therapeutic options.

Clinical implications

The high prevalence of patients with chronic pain, and especially with NSCNP, makes it one of the most relevant musculoskeletal disorders in the rehabilitation sciences (Vos et al., 2013). It is therefore essential to develop new approaches to rehabilitation strategies. In this regard, motor control exercises have been shown to decrease pain and disability in patients with NSCNP compared with other types of treatment (Martin-Gomez et al., 2019). However, the clinical implementation of this type of exercise in a clinical context is challenging, due to its high complexity or the pain itself leading patients to not perform it or to perform it incorrectly, reducing its therapeutic potential. In this regard, both MI and AO provide a simple, clinically therapeutic alternative at low cost that can be performed independently by the patient. The results of this study suggest that mental practice can be a useful therapeutic strategy in patients with NSCNP, especially in the early stages of rehabilitation, and both strategies should be considered for patients with NSCNP.

Limitations

This study presents several limitations. First, the sample size is small; thus, the results should be considered with caution. In addition, the results have only been considered in the short term, and the duration and type of intervention might be insufficient for greater increases in CJPS in patients with NSCNP. Hinshaw (1991) have found that the optimal time for MI to provide the greatest benefits is between 10 and 15 min. In the present study, the duration of the MI intervention was 4 min. This length of time might not be sufficient to obtain the full potential of MI. Further research is needed to determine the role of mental practice in the rehabilitation process of patients with NSCNP.

Conclusions

The results obtained in the present study showed that the AO group obtained greater improvements than the PO group in CJPS for the cervical extension movement both at the end of the intervention and 10 min postintervention, as well as in the cervical flexion movement, although only at 10 min postintervention. In addition, the AO group obtained greater improvements than the MI group in the CJPS only at the end of the intervention in the cervical extension movement. Finally, regarding the left rotation cervical movement, both MI and AO were superior to PO in both planes at the end of the intervention.

Our results suggest that AO training is an effective sensorimotor neurotraining tool to improve CJPS in the early stages of treatment. In addition, MI could also be a tool to consider using in rehabilitation, but perhaps with a longer training time. The therapeutic potential of the application of mental practice in a clinical context in the early stages of rehabilitation of NSCNP should be considered.

Supplemental Information

Supplemental Information 1 Raw data

Click here for additional data file.

Supplemental Information 2 Consort checklist

Click here for additional data file.

Supplemental Information 3 Trial protocol

Click here for additional data file.

Additional Information and Declarations

Competing Interests

Author Contributions

Clinical Trial Ethics

Data Availability

Clinical Trial Registration

The authors declare there are no competing interests.

Ferran Cuenca-Martínez conceived and designed the experiments, performed the experiments, analyzed the data, contributed reagents/materials/analysis tools, prepared figures and/or tables, authored or reviewed drafts of the paper, approved the final draft.

Roy La Touche conceived and designed the experiments, analyzed the data, contributed reagents/materials/analysis tools, authored or reviewed drafts of the paper, approved the final draft.

Jose Vicente León-Hernández analyzed the data, authored or reviewed drafts of the paper, approved the final draft.

Luis Suso-Martí conceived and designed the experiments, performed the experiments, analyzed the data, authored or reviewed drafts of the paper, approved the final draft.

The following information was supplied relating to ethical approvals (i.e., approving body and any reference numbers):

La Salle University Center for Higher Education granted Ethical approval to carry out the present study (Ethical Application Ref: 027/2019).

The following information was supplied regarding data availability:

The raw measurements are available in Supplemental File.

The following information was supplied regarding Clinical Trial registration:

The trial registry number is NCT03910829.

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
