# Peer review of "Mental practice in isolation improves cervical joint position sense in patients with chronic neck pain: a randomized single-blind placebo trial"

_PeerJ, doi:10.7717/peerj.7681_

## Round 0.1 · original submission · Minor Revisions

The reviewers and I see considerable potential in this manuscript and therefore would like you to resubmit based on their suggestions.

Reviewer 1 ·

Basic reporting

ok

Experimental design

ok

Validity of the findings

ok

Additional comments

TITLE
1. ok.
ABSTRACT
1. ok

INTRODUCTION
2. ok
METHODS.
3. okhas been used.
RESULT
4. ok
DISCUSSION
5. could be re-written to describe in brief.
Tables
6. ok

Reviewer 2 ·

Basic reporting

The English language is appropiate, the background and literature references are correct, figures and tables are very useful, raw data is provided and relecant results are presented.

Experimental design

The aims are with the scope of the journal, the research question is well defined. Please, see comments to authors in order to improve this manuscript. Ehics information is added. Methods are well described.

Validity of the findings

The validity of findings is well described. Conclusion are supported by the study findings.

Additional comments

Thanks for the opportunity to review this interesting manuscript. This is a well designed clinical trial that provided importants considerations for the biomedical field. Please, consider the following issues/comments according to the presented manuscript:

ABSTRACT
-Please, consider add subheadings to the abstract.

INTRODUCTION
-Paragraph from line 40 to 55: Please, a phrase linking sentral sentization to motor control alterations would be useful at the end of tis paragraph (please, reference this paargraph appropiately).

METHODS
-The study was adequately designed and registered prospectively at clinicaltrials.gov.
-References to support inclusion and exclusion criteria would be useful. In addition, the Declaration of helsinki should be cited.
-Randomization, blinding and interventions were well-detailed.
-Reliability coefficients and minimum detectable changes were adequately provided for the primary outcome.
-Reliability coefficients need to be provided for some secondary outcomes tools such as Tampa scale, NDI and level of physical activity.
-A sample size calculation is lacking. Please, provide this calculation or if it is not possible, please, include this issue in limitations and detail this manuscript as a pilot study.

RESULTS
-Tables and figures were well presented and provided useful information.

DISCUSSION
-Please, consider to dicuss about some future studies comparing AO and MI with other interventions such as dry needling or ischemic compression of myofascial trigger points (Acupunct Med. 2019 Jun;37(3):151-163. doi: 10.1177/0964528419843913) and mobilization of the upper cervical region (J Oral Rehabil. 2019 Feb;46(2):109-119. doi: 10.1111/joor.12733) to improve cervical motor control in subjects with non-specific neck pain.

·

Basic reporting

well written paper covering an interesting area of physiotherapy.

Experimental design

No comment

Validity of the findings

No comments

Additional comments

Please see attached comments.

---

## Round 0.2 · accepted · Accept

We would like to thank the authors to attending to our comments.

Reviewer 2 ·

Basic reporting

Ok

Experimental design

Ok

Validity of the findings

Ok

Additional comments

I congratulate to the authors due to the manuscript has widely improved and the authors have responded all my prior comments. Thanks for your contribution to the physical therapy field.

·

Basic reporting

The basic reporting is clear and professional English is used throughout.

Experimental design

Good experimental design.

Validity of the findings

Conclusions and supported by research findings.

Additional comments

The authors have revised the manuscript very well and have attended to all comments made.
Well done on this very interesting piece of work. Well done on providing very clear clinical implications.